# Seismic resolution improving by a sequential convolutional neural network

Zhenyu Yuan[1,2], Yuxin Jiang[3], Zheli An[1,2,4]*, Weibin Ma[1,2], Yong Wang[1,2]

1 Railway Engineering Research Institute, China Academy of Railway Sciences Corporation Limited, Beijing, China, 2 State Key Laboratory of High-Speed Railway Track System, Beijing, China, 3 PST Service Corporation, Beijing, China, 4 China Academy of Railway Sciences, Beijing, China

* anzheli95@qq.com

## Abstract

Thin-bed soft rock is one of the main factors causing large deformations of tunnels. In addition to relying on some innovative construction techniques, detecting thin beds early during surface geological exploration and advanced geological prediction can provide a basis for planning and implementing effective coping measures. The commonly used seismic methods cannot meet the requirement for thin beds detection accuracy. A high-resolution (HR) seismic signal processing method is proposed by introducing a sequential convolutional neural network (SCNN). The deep learning dataset including low-resolution (LR) and HR seismic is firstly prepared through forward modeling. Then, a one-dimension (1D) SCNN architecture is proposed to establish the mapping relationship between LR and HR sequences. Training on the prepared dataset, the HR seismic processing model with high accuracy is achieved and applied to some practical seismic data. The applications on both poststack and prestack seismic data demonstrate that the trained HR processing model can effectively improve the seismic resolution and restore the high-frequency seismic energy so that to recognize the thin-bed rocks.

**Data Availability Statement:** This manuscript's minimal data set is publicly available. The code is shared openly on Github (https://github.com/seisgo/SequenceSuperResolution).

## 1. Introduction

Driven by the need for economic development, China's transportation sector is booming, as reflected by the increasing number of ongoing construction projects related to transportation infrastructure. In western China, where mountains and plateaus are dominant landforms, the number of tunnels under construction has increased significantly. In recent years, the problem of large deformation caused by soft rock is often encountered at the construction sites of mountain tunnels, which negatively aggravates safety risks, increases construction costs, and extends the construction period [1, 2]. Squeezed surrounding rocks, primarily thin beds of rock or rock sheets [3], are the main factor causing large deformations of tunnels [4–8]. In order to solve the problem of large deformation, one approach is to innovate construction techniques, and another is to identify the high-risk area of large deformation early and take coping measures, including modifying design, taking preventive measures or directly avoiding the high-risk area. The second approach is usually more economical than the first one.

**Funding:** This work was supported in part by the National Natural Science Foundation of China High-speed Rail Joint Fund under Grant U1934218. The funders had no role in study design, data collection and analysis, decision to publish, or preparation of the manuscript.

**Competing interests:** The authors have declared that no competing interests exist.

Seismic exploration is a type of important technique to identify the unfavorable geology in advance. The seismic resolution is a metric to measure the ability to distinguish two neighboring strata from seismic signals. The rich information in HR seismic data is conducive to in-depth exploration of underground geological targets. During wave propagation, each stratum functions like a band-pass filter, the gathered seismic signals are band-limited and the high-frequency components attenuate rapidly with depth. The attenuation of the seismic signal is manifested by narrowing the frequency band and dropping the dominant frequency, decreasing the resolution of seismic signals. When the strata are thin, the reflected waves of neighboring interfaces overlap with each other therefore are challenging to distinguish. The phenomenon of seismic attenuation is particularly detrimental to the capability of identifying thin-bed rock. In order to better recognize underground geological bodies such as thin beds, HR seismic processing is an essential part of the work.

Based on signal theory, scholars have adopted or developed techniques such as deconvolution, inverse Q filtering [9, 10], and frequency recovery to improve seismic resolution. Sajid and Ghosh [11] employed a set of three cascaded difference operators to boost high frequencies and combined with a simple smoothing operator to boost low frequencies. Wang et al. [12] proposed an adaptive spectrum broadening method based on the molecular-Gabor transform to improve the resolution of nonstationary seismic data. Kahoo and Gholtashi [13] fully utilized the valid information of the spectrum and cepstrum based on logarithmic time-frequency transform to extend the frequency band at each translation of the spectral decomposing window. Chen and Wang [14] proposed a wavelet compression method that utilizes the scale characteristic in the Fourier transform. Mohamed [15] compared negative of the second derivative with band-pass filter and spectral bluing, demonstrating that negative of the second derivative is an effective enhancing-frequency technique which enhances the resolution while maintaining the poststack seismic characteristics such as zero-phase and polarity. It is remarkable that frequency enhancement does not guarantee vertical resolution improvement [16]. The above approaches usually rely on some specific assumptions, limiting their applications.

In recent years, the rapid development of big data, high-performance computing, and artificial intelligence has promoted the application of data-based deep learning methods in HR seismic processing. Yuan et al. [17] constructed a regression model by means of support vector machine that can be used for HR seismic processing when the target curve is an HR seismic trace. However, such method treats seismic signals discretely so that the consequence of spatial correlation and structural features of the target strata is ignored. Convolutional neural network (CNN) can effectively mine the correlation features of structured data through convolution operations and achieves a big achievement in image processing fields, such as image denoising and super-resolution processing. Therefore, CNN has been introduced into the field of seismic image enhancement. Wang and Nealon [18] applied a U-Net CNN for seismic image enhancement to better reconstruct geological structures and simplify interpretation. Considering the limitations of preparing HR seismic data samples, Halpert [19] trained a generative adversarial network to generate HR seismic images based on a small training dataset and achieved good results. Preparing input and target samples by forward modeling, Yuan et al. [20] proposed a 1D CNN architecture to perform HR seismic processing.

Inspired by the impressive achievements of HR seismic processing deep learning, we develop a seismic resolution improving method by a SCNN architecture, which is suitable for thin beds detection. The rest of the paper is arranged as follows. We first present the architecture and some basic configurations of the SCNN. Then, a training dataset is prepared through forward modeling. Afterwards, the SCNN model is trained on the prepared dataset and applied to practical poststack and prestack seismic data to manifest its performance. Finally, we draw some conclusions.

## 2. Sequential convolutional neural network

Considering that seismic trace is essentially a type of time series, 1D LR and HR seismic traces are used as the deep learning network's input and output. The neural network model for HR seismic processing developed in this study is called a sequential convolutional neural network (SCNN), whose architecture is a type of encoder-decoder network, with additional residual connections, shown as in Fig 1. The architecture was firstly developed by Kuleshov et al. [21] to perform audio super-resolution, which involves upscaling of time series. Referenced here, upscaling operation is eliminated since the sampling number of both input and output seismic trace are same.

The SCNN model utilizes a 1D convolutional network to process time series. It contains symmetric downsampling and upsampling blocks. Each block performs a convolution, batch normalization and a nonlinear activation. At a downsampling step, we halve the time dimension and double the filter size. During upsampling, this is reversed. Downsampling is performed by a 2-stride convolution or max pooling. Upsampling is performed by a 1-stride convolution and a 1D version of Subpixel layer, named SubPixel1D [21]. The SubPixel1D layer maps an input tensor of dimension $f{\times}c$ into one of size $2f{\times}c/2$, which has been shown to be less prone to produce artifacts than transposed convolution [22]. In addition, a bottleneck block composed of convolution connects the downsampling and upsampling blocks.

During high-resolution processing, the input LR seismic trace is closely related to the target HR seismic trace. In spite of the same low-frequency trend, attributes including amplitude, phase, and waveform, are closely correlated, which can be well extracted by the encoder-decoder architecture. Therefore, when employing an SCNN model to establish the relationship between HR and LR seismic traces, the output feature of each convolution in the downsampling process contributes to the symmetric convolution operation in the upsampling process [23]. After proper convolution operations, the outputs of the symmetric convolution layers of the downsampling and upsampling blocks have same matrix dimension and are correlated. On this basis, connecting the symmetric matrices with residual connections can effectively

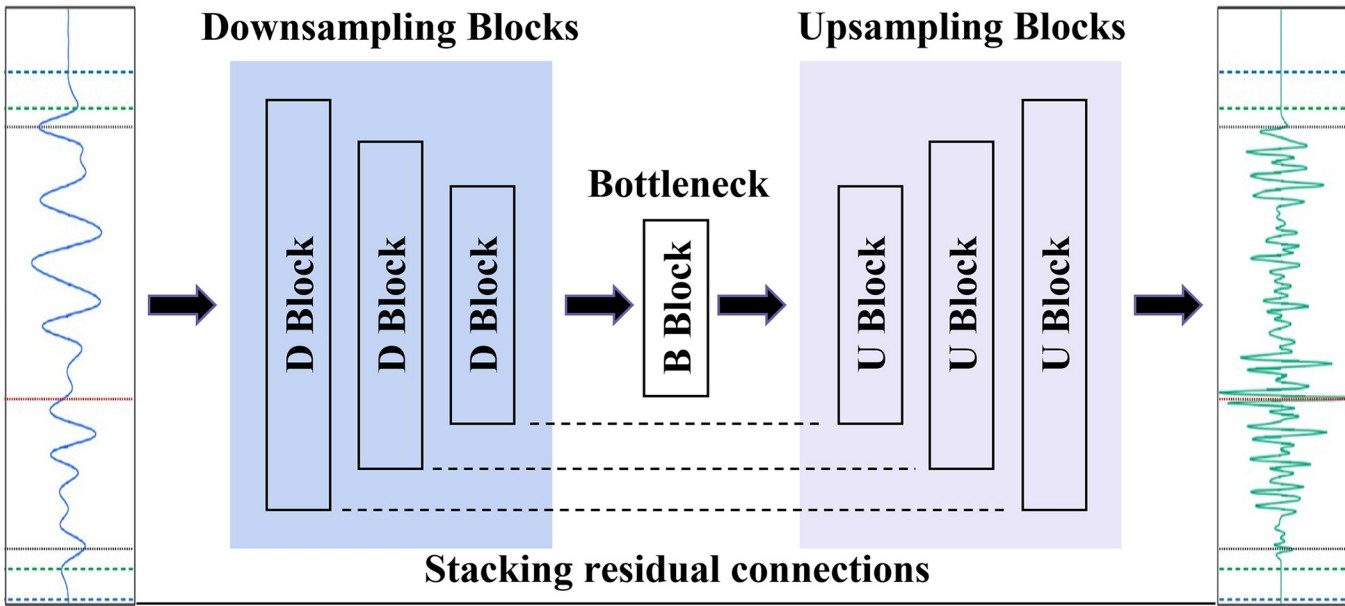

**Fig 1. Architecture of SCNN for HR seismic processing.**

establish a nonlinear relationship between the LR input data and the HR output data. Residual connections contribute to speed up the model training and reduce risk of overfitting, thus improve the efficiency and accuracy of the optimal solution solving [24].

In summary, the encoder-decoder architecture allows SCNN to extract high-hierarchy features such as waveform, amplitude, frequency and phase from seismic trace, further establishing the correlation between HR and LR seismic signals. Residual learning relying on the residual connections can promote adequate training and learning of the neural network model, ensuring the efficiency and accuracy of optimal solutions. Therefore, SCNN can be effectively applied to the HR processing of seismic data.

The code is shared openly on Github (https://github.com/seisgo/SequenceSuperResolution).

## 3. Data preparation

In addition to the network architecture, training dataset plays an important role to the performance of machine learning. For data-driven HR seismic processing, both LR and HR seismic trace are required. Gathered band-limited seismic can be used to provide LR data, however the corresponding real HR seismic is unknown. An approximate HR counterpart for the LR input can be achieved by commonly used theory-driven HR processing techniques, which may lack wide suitability.

### 3.1. Wavelet selection

Well logging provides high-resolution and relatively accurate information of reservoir formations, where acoustic and density logging curves are used to provide log reflectivity. To prepare correlative features and labels, synthetic seismic traces are then synthetized by convolution from log reflectivity with LR and HR wavelets. The selection of wavelet is significant, especially for the synthesis of HR seismic traces. It is desirable that the wavelet is near sharp pulse in the time domain and correspondingly as wide a band as possible in the frequency domain, so that the influence of side lobes can be minimized. Ricker wavelet is a commonly used theoretical wavelet with large side lobes, the magnitude of its dominant frequency affects its resolution. Wide-band Ricker wavelet, as a integration of multiple Ricker wavelets of different bandwidths, has small sidelobes and is superior to the Ricker wavelet in terms of fidelity and signal-to-noise ratio, expressed as

$$y(t) = \frac{q e^{-(\pi q t)^2} - p e^{-(\pi p t)^2}}{q - p}, \tag{1}$$

where $p$ and $q$ are the lower and upper limits of the integration function, controlling the shape and bandwidth of the wide-band Ricker wavelet.

Cao et al. [25] proposed a four-parameter wide-band B-spline wavelet based on the wavelet decomposition theory of seismic pulse and its reconstruction.

$$y(t) = \frac{q\, sinc(2qt) - p\, sinc(2pt)}{q - p} \sqrt{f_b}\, sinc\left(\frac{f_b t}{m}\right)^m, \tag{2}$$

where $f_b$ is the bandwidth, $m$ is an integer adjustment parameter. Controlled by parameter $f_b$, $m$, $p$ and $q$, the wide-band B-spline wavelet allows the adjustment of the attenuation of side lobes and the number of phases.

The Ricker wavelet, wide-band Ricker wavelet and wide-band B-spline wavelet are compared in Fig 2. The dominant frequency of the Ricker wavelet is 30 Hz, lower and upper limits

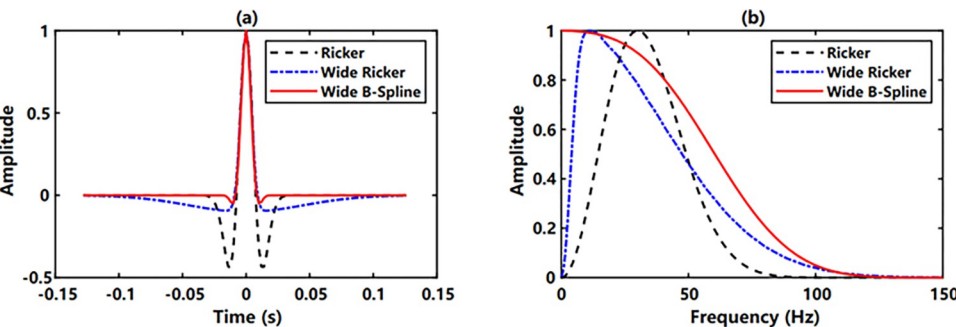

**Fig 2.** Comparison of different wavelets: (a) Time domain (b) Frequency Domain.

of the wide-band Ricker wavelet are 5 and 55 Hz respectively, $m$ value of the wide-band B-spline wavelet is 5, and the bandwidth $f_b$ is 200 Hz. It demonstrates that with almost same main lobe width in time domain, the wide-band B-spline wavelet has the smallest side lobe and the shortest non-zero continuation time, and the wide-band Ricker wavelet has smaller side lobes than the Ricker wavelet but has a longer non-zero continuation time, which leads to truncation in applications [19]. It can be observed from the amplitude spectra that the wide-band B-spline wavelet maintains the low frequency very well and has the widest bandwidth, while the dominant frequency of the wide-band Ricker wavelet considerably decreases when there is no significant increase in bandwidth. In summary, the wide-band B-spline wavelet exhibits the best ability for object distinction.

Given a randomly generated reflection coefficient sequence (Fig 3) and convolved with the above wavelets, three seismic traces are synthesized and corresponding amplitude spectra are calculated, shown in Fig 4. From Figs 3 and 4A, we know that the wide-band Ricker wavelet and the wide-band B-spline wavelet represent the variation of the reflection strength very well. However, large side lobes of the Ricker wavelet contribute to the amplitude accumulation of the adjacent opposite-direction reflections, creating a fake strong reflection. Moreover, the long non-zero continuation time of the wide-band Ricker wavelet introduces time delay of adjacent codirectional reflection waveforms, thus reducing the seismic resolution. From the spectra (Fig 4B), the wide-band B-spline wavelet further demonstrates widest frequency bandwidth and best energy recovery in both low and high-frequency segments.

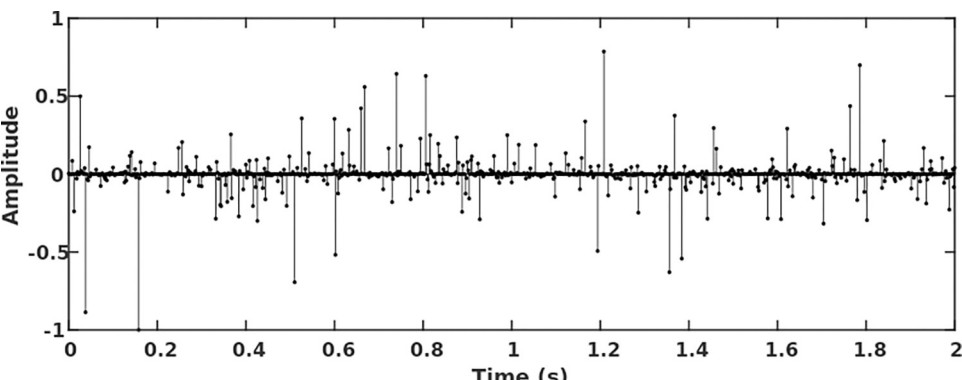

**Fig 3. Generated seismic reflection coefficient sequence.**

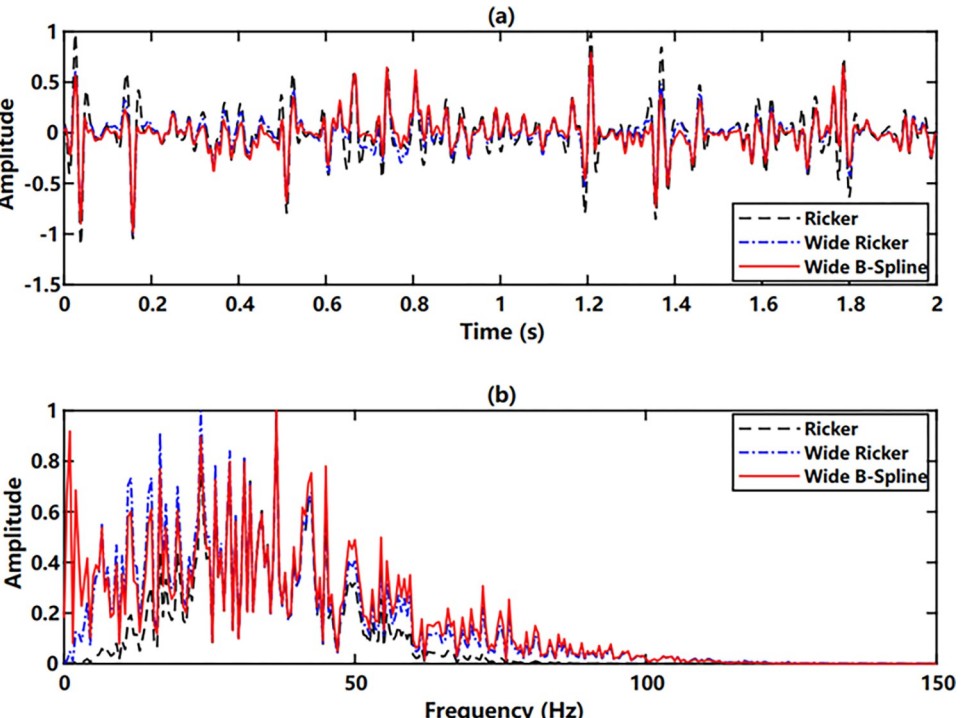

**Fig 4.** Synthetic seismic traces and amplitude spectra by different wavelets: (a) synthetic seismic traces (b) amplitude spectra.

## 3.2. Dataset generation

Practical acoustic and density logging curves from 14 wells are chosen to calculate log reflectivity, shown in Fig 5A. Corresponding LR and HR synthetic seismic traces (S1 and S2 Files) are derived by convolving the reflectivity with Ricker wavelet and wide-band Ricker wavelet with dominant frequency of 30 Hz separately, shown in Fig 5B and 5C. Maintaining same phase and amplitude information, the HR synthetic seismic demonstrates higher temporal resolution than the LR synthetic seismic.

Fig 6 further zooms in the reflections from 300 to 600 ms for the 4th well. The reflectivity curve is indicated as black sticks, showing the location of reflection interfaces as well as the polarity and magnitude. The solid blue and red lines represent the LR and HR synthetic seismic, respectively. The LR synthetic seismic cannot distinguish the reflection waveforms of neighboring interfaces, only the envelope of the reflection waveforms can be observed, which is inconsistent with the phase and amplitude of the actual reflectivity curve. By contrast, the phase and amplitude of the HR synthetic seismic are consistent with the reflection coefficient, indicating that the HR synthetic seismic distinguishes the main reflection interfaces well. Convolving with same reflectivity curve, the HR and LR synthetic seismic are inherent correlated and have similar low-frequency trend, thus provide available training samples for HR processing deep learning.

In addition, synthetic data samples can be prepared based on some typical geological models, which demonstrate the practical variation of subsurface geological and geophysical properties. Marmousi2 model [26] is a classic geological model commonly adopted to study seismic migration and fluid identification, expressing complex geological structures, lithologies, and fluids. The P wave impedance of the Marmousi2 model is calculated from the P wave velocity and density model, shown in Fig 7. Further, 30 Hz Ricker wavelet and wide-band B-spline

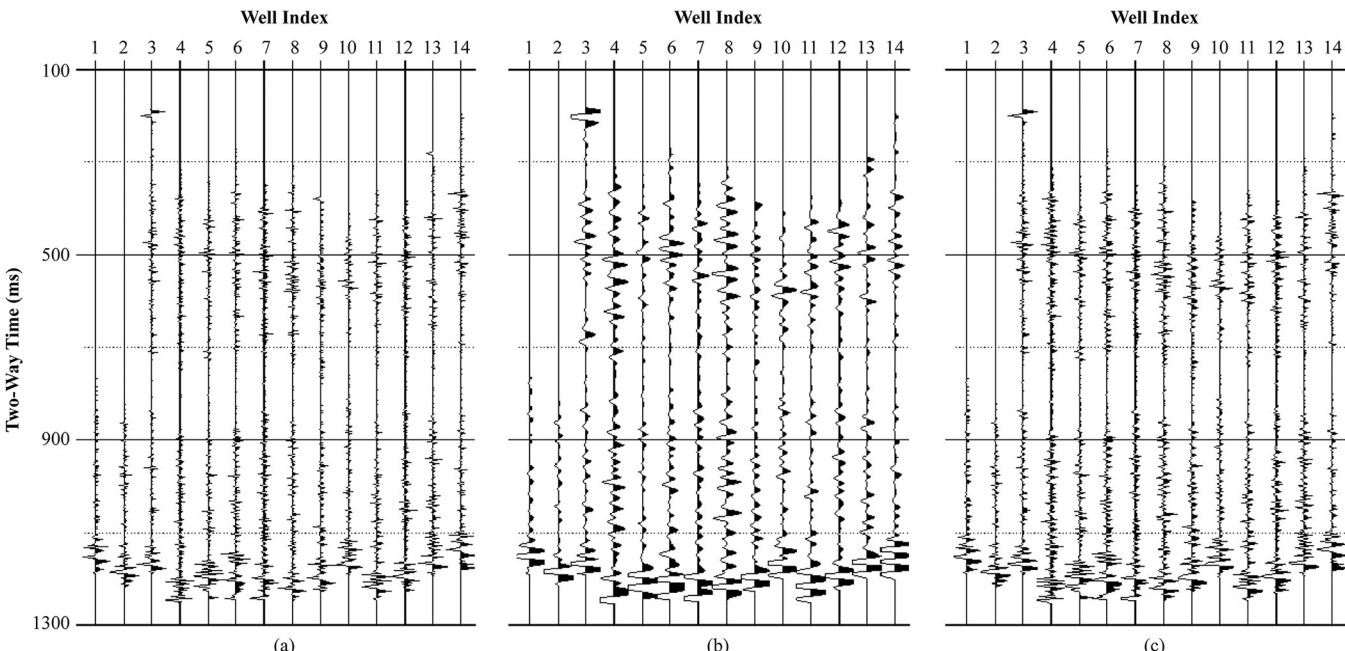

**Fig 5.** Seismic synthetics by convolution from 14 wells: (a) log reflectivity (b) synthetic traces with a 30Hz Richer wavelet (c) synthetic traces with a wide-band Ricker wavelet.

wavelet are separately utilized to synthetize LR (Fig 8) and HR (Fig 9) seismic data (S3 and S4 Files). The HR synthetic seismic displays stratigraphic interfaces well and even clearly depict some fluid interfaces. In comparison, some thin beds are concealed in the envelopes of adjacent waveforms for the LR synthetic seismic.

The Marmousi2 model contains more than 13,000 traces, extracting seismic traces from its synthetic seismic at a specific interval of 20 is available to achieve a large dataset for HR seismic processing deep learning.

Some preprocessing steps are performed to prepare the proper samples for deep learning. Considering the amplitude difference of different seismic traces, all samples are firstly scaled to a uniform numerical range with maximum absolute value of 1. Then, the LR and HR seismic traces are segmented into samples with same time length (such as 512 sampling points). Finally, the dataset (S5 File) is randomly split into training, validation, and test dataset at a certain ratio for model training, validation, and evaluation.

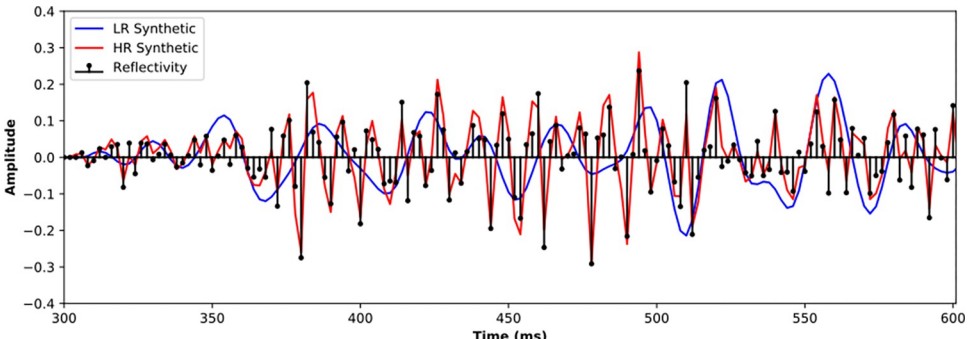

**Fig 6. Synthetic LR and HR seismic traces of a certain well.**

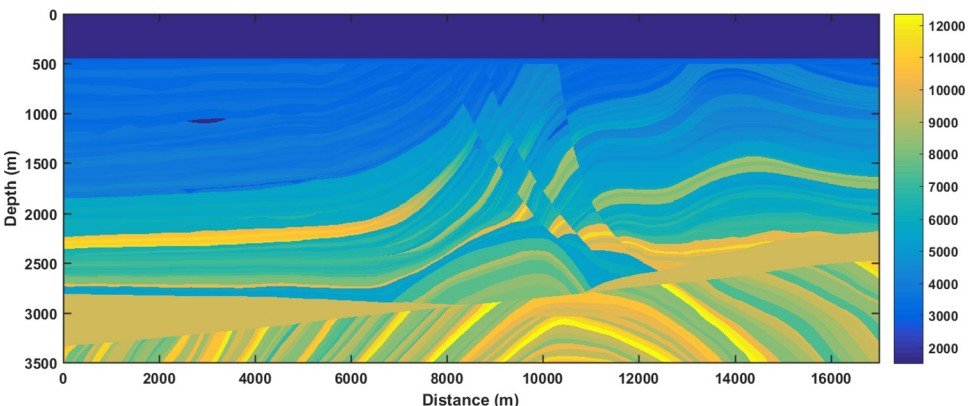

**Fig 7. P wave impedance of Marmousi2 model.**

## 4. Model establishment

### 4.1. Model training

Indicating the source and target seismic trace as $\mathbf{X}$ and $\mathbf{Y}$ separately, the nonlinear transformation $F$ between $\mathbf{X}$ and $\mathbf{Y}$ is learned by the SCNN model, expressed as

$$\mathbf{Y} = F(\mathbf{X}), \tag{3}$$

where $F$ is an abstract expression of the SCNN, determined by operations including convolution, batch normalization, activation and others.

To solve this nonlinear function, mean squared error (MSE) objective function is adopted, expressed as

$$MSE = \frac{1}{M} \sum_{i=1}^{M} \| F(\mathbf{X}_i) - \mathbf{Y}_i \|^2, \tag{4}$$

where $i$ indicates the instance index of machine learning dataset $\{\mathbf{X}_i, \mathbf{Y}_i\}$, varying from 1 to $M$.

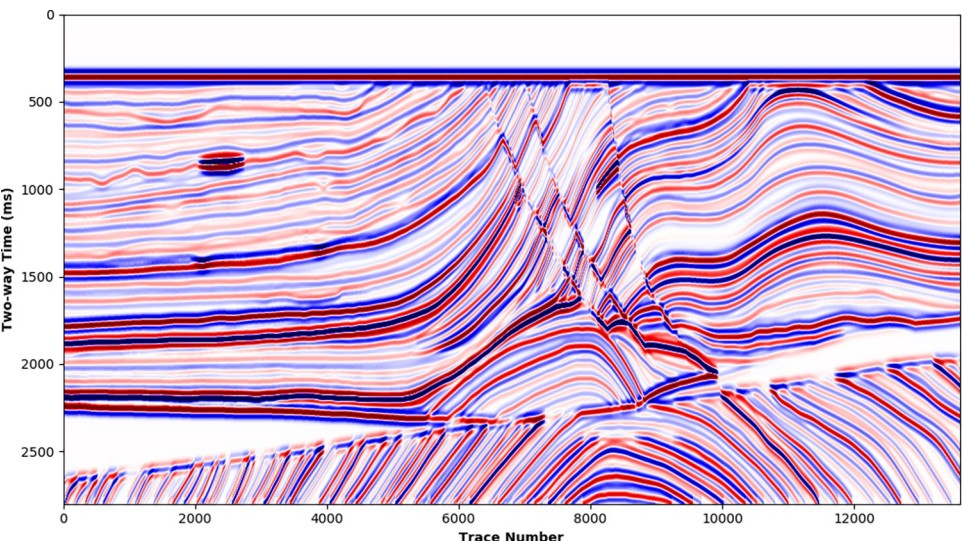

**Fig 8. Low-resolution synthetic seismic of Marmousi2 model convoluted with a 30Hz Richer wavelet.**

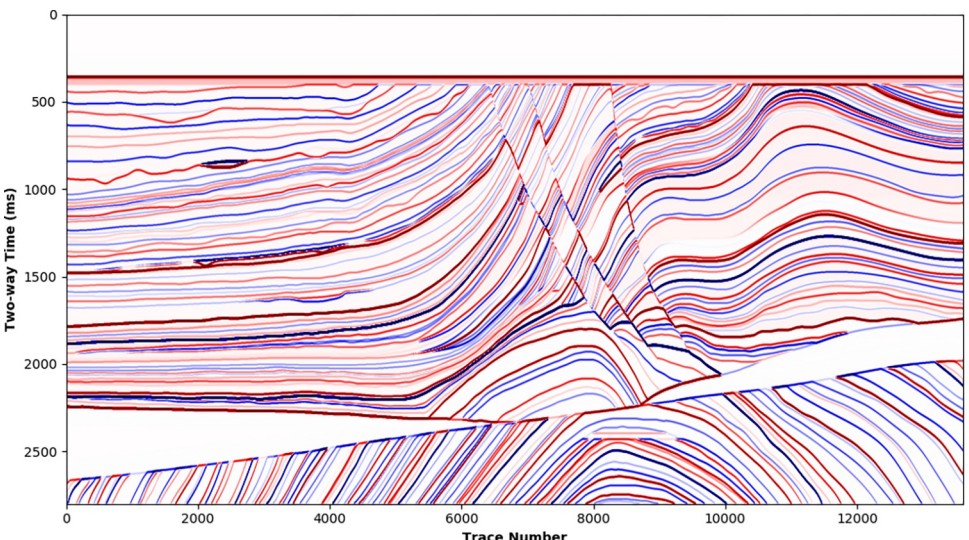

**Fig 9. High-resolution synthetic seismic of Marmousi2 model convoluted with a wide-band B-spline wavelet.**

The Adam optimizer is adopted to perform optimization. Fig 10 shows the training performance and validation performance, indicated by the blue and orange curve. At the early iteration stage, training and validation errors decrease rapidly. As the iteration progresses, the loss curve gradually smooths, and the model optimization is thus completed.

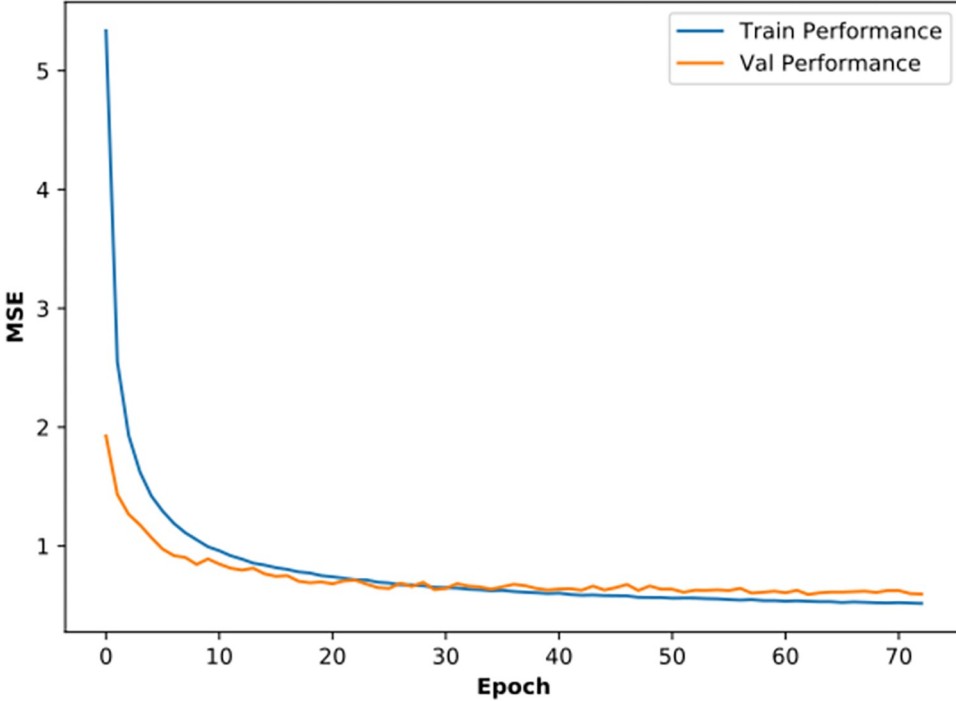

**Fig 10. Training performance of high-resolution seismic processing SCNN model.**

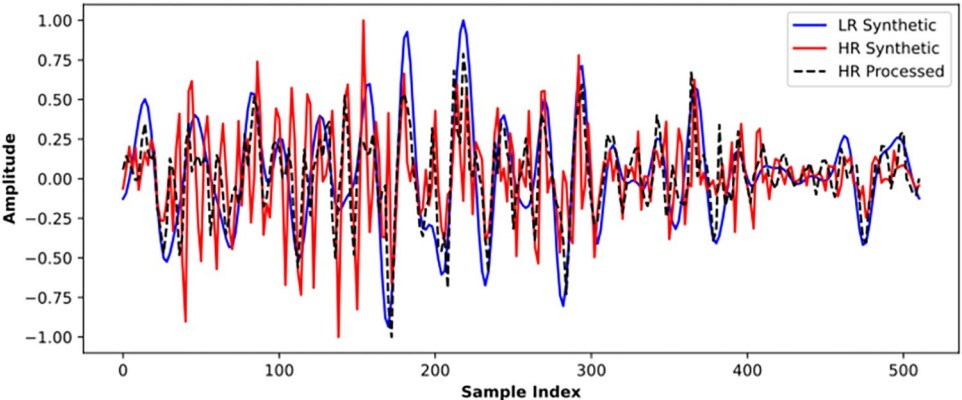

**Fig 11. Validation of the well-trained SCNN model by well synthetic data.**

## 4.2. Model evaluation

The well-trained model is applied to the test dataset to evaluate its prediction performance on unknown samples. Fig 11 shows an application on a logging curve. The HR synthetic depicts the transition of the interface, whereas the LR synthetic is so smooth that cannot distinguish many small reflection interfaces. The HR processed result recovers some sharp pulse changes and is more consistent with the HR synthetic. In particular, at sample index 100, the missed positive reflection in the LR synthetic is recovered in the HR processing result.

Fig 12 further shows an application on a certain trace of the Marmousi2 model. For the LR synthetic trace, the main lobe of the reflection wave is wide, and the side lobes are large and have a long-time delay, so the time resolution is very low. In comparison, the HR processed result is almost identical to the HR synthetic trace, indicating a significant resolution improvement.

In summary, the SCNN-based HR processing model trained on synthetic datasets is of high accuracy and can effectively improve the resolution of seismic data, which is suitable for processing actual seismic data.

## 5. HR processing applications

### 5.1. Poststack HR processing

For the Marmousi2 model whose most traces not included in the training dataset, the SCNN model is applied to the LR synthetic to get the HR processed result (S6 File), shown in Fig 13. In comparison with the LR synthetic (Fig 8), the HR processed result significantly improves

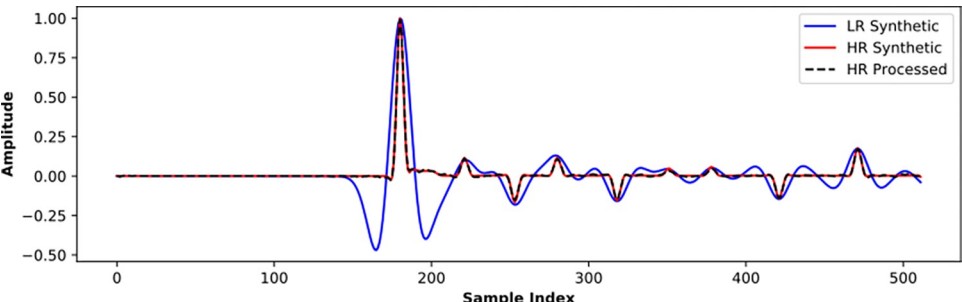

**Fig 12. Validation of the well-trained SCNN model by Marmousi2 synthetic data.**

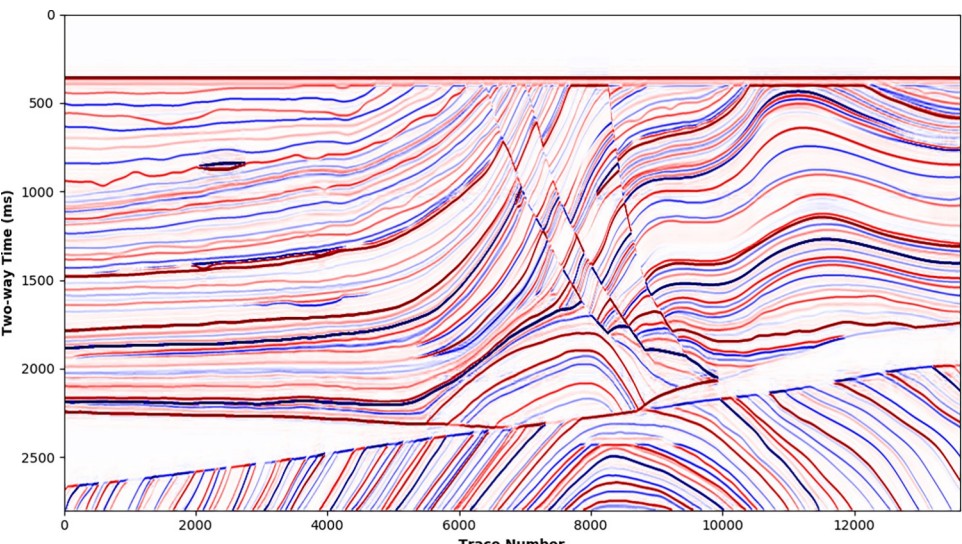

**Fig 13. High-resolution processing for Marmousi2 low-resolution synthetic by SCNN model.**

the seismic resolution and recovers thin-bed reflections, which is consistent with the HR synthetic shown in Fig 9. Fig 14 further compares single traces in the HR synthetic and HR processed result. Despite some slight amplitude and phase deviations at adjacent weak reflection interfaces (2000~2100 ms), the HR processed result is in good agreement with the HR synthetic.

The trained SCNN model is then applied to a real poststack seismic data (Fig 15). After HR processing, some weak events are recovered and some overlapped adjacent events are distinguished, and the reflection energy are more balanced. In particular, the strong reflection events at 1100 and 1360 ms suppress neighboring reflections for the raw seismic, the HR processed seismic effectively restore neighboring weak reflections and match the wellbore synthetic seismic trace (red curve) well.

The near-well trace is further analyzed in time domain (Fig 16) and frequency domain (Fig 17). It is noticed that the SCNN HR processing effectively improves seismic resolution. For seismic trace in Fig 16, more weak reflections are enhanced and some overlapped reflections are restored. For amplitude spectrum in Fig 17, the frequency band is broadened and the high-

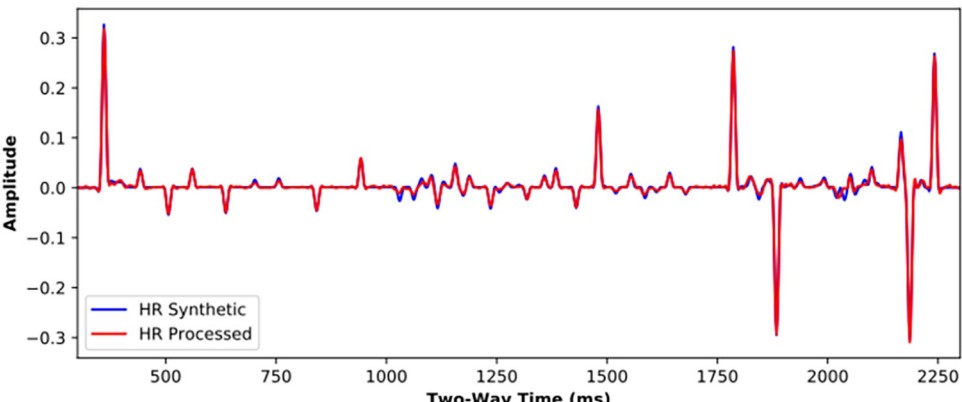

**Fig 14. Single trace comparison for high-resolution processing of Marmousi2 model.**

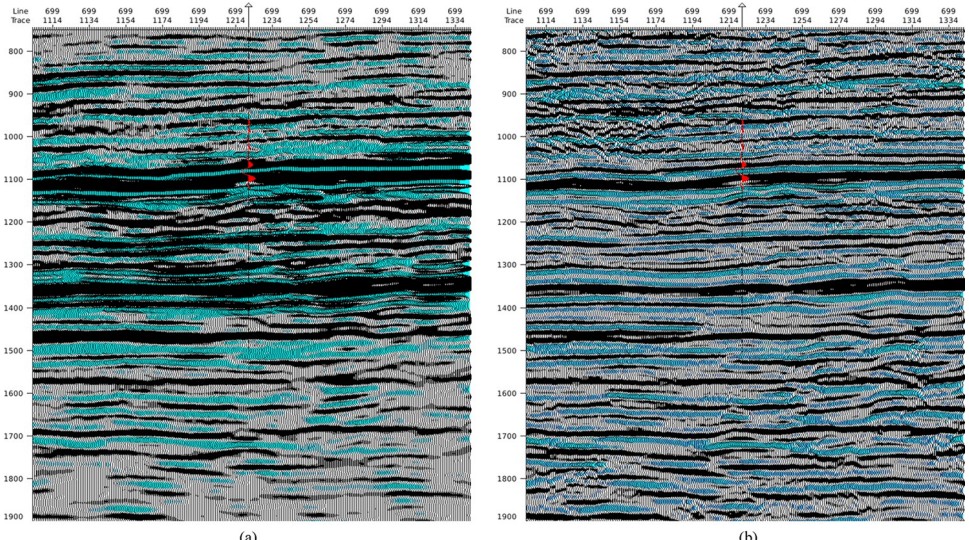

(a)　　　　　　　　　　　　　　　　　(b)

**Fig 15.** Processing to a poststack seismic data by the well-trained SCNN model: (a) raw seismic section (b) HR processed seismic section.

frequency energy is restored. The HR processed seismic is conducive to discerning the details of stratigraphic sedimentation.

## 5.2. Prestack HR processing

The trained SCNN HR processing model is applied to a prestack common depth point (CDP) gather. Figs 18 and 19 display the raw and HR-processed CDP gathers. After HR processing, the seismic resolution is effectively improved, making the weak reflections adjacent to strong reflections identified effectively. Consequently, the previously weak and unclear reflection events are restored, and deep reflections are enhanced. In addition, HR processing enhances the near-offset reflection energy, indicating a good energy balance.

Figs 20 and 21 further compare the near-well raw and HR-processed seismic traces and their corresponding amplitude spectra. Similar to the poststack HR processing, the HR processing SCNN model effectively improves seismic resolution. The frequency band is effectively

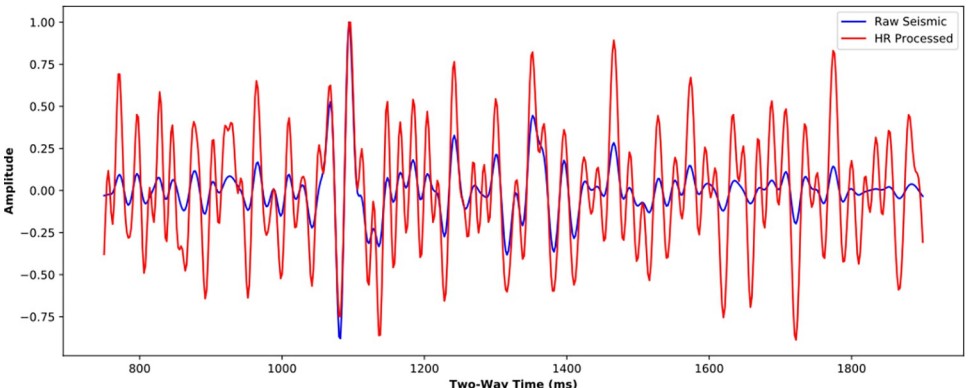

**Fig 16. Comparison of near-well raw trace and its corresponding HR processed trace from the poststack seismic data.**

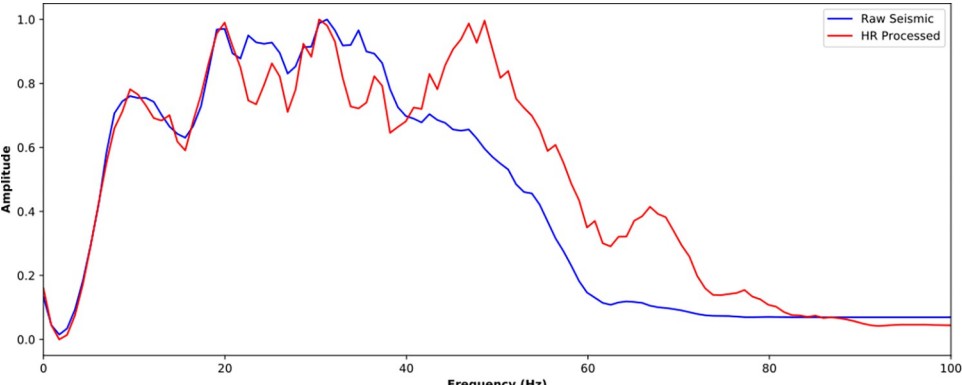

**Fig 17. Spectra comparison of near-well raw trace and its corresponding HR processed trace from the poststack seismic data.**

broadened with a rate of 41.06%, the high-frequency energy is restored, and the low-frequency energy is compensated at a certain extent. The convergence of reflection waveforms is enhanced, and some reflections of thin beds are highlighted, which is conducive to performing AVO inversion and improving anisotropic fracture detection.

## 6. Conclusions

Taking 1D LR and HR seismic trace as input and output, a data-driven deep-learning-based method is developed to improve seismic resolution. As a sequential convolutional neural network, the proposed method, taking advantage of encoder-decoder network and residual learning, is appropriate for learning complex transformation between LR and HR trace pairs. Besides network architecture, training dataset influences the learning performance significantly. Considering the rich stratigraphic information contained in the logging curves and typical geological models, appropriate LR and HR wavelets are selected to generate adequate LR-HR dataset by forward modeling. The HR processing model is well trained and shows

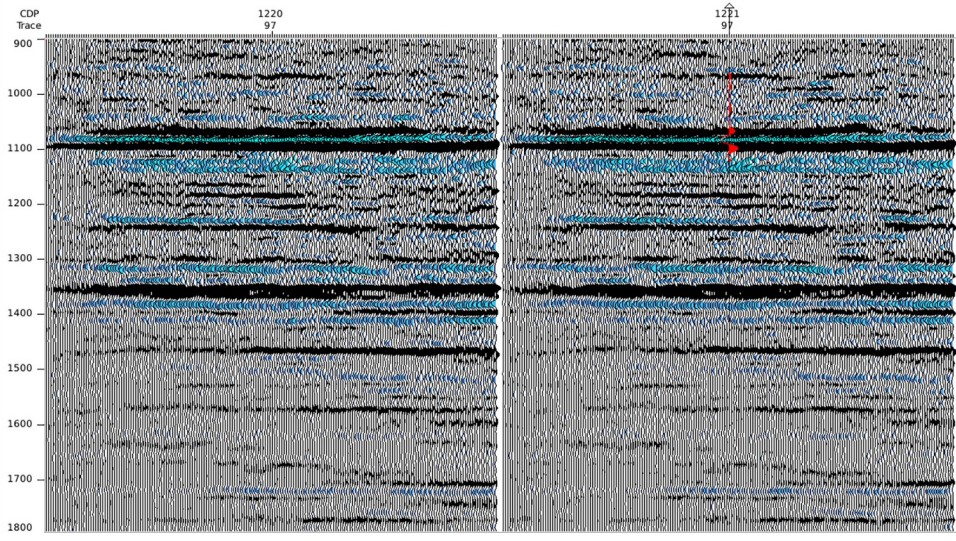

**Fig 18. Raw seismic gathers.**

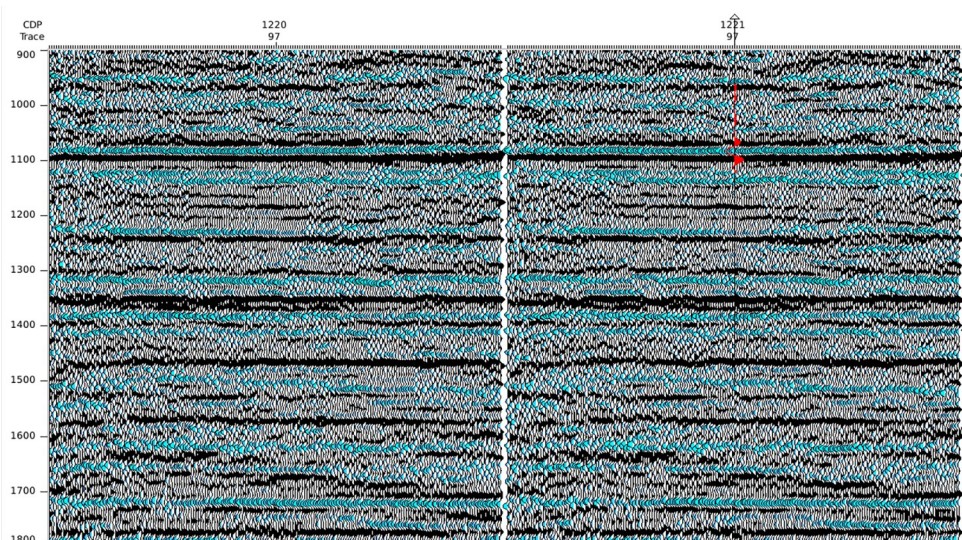

**Fig 19. HR processed seismic gathers by the well-trained SCNN model.**

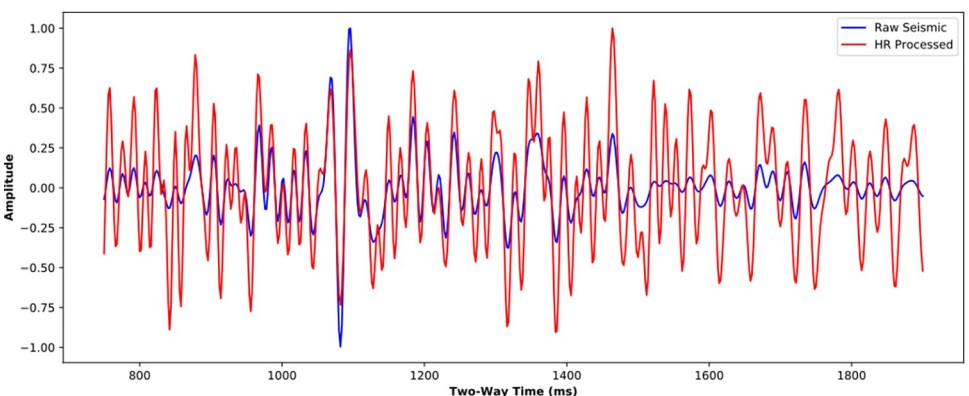

**Fig 20. Comparison of near-well raw trace and its corresponding HR processed trace from the prestack seismic data.**

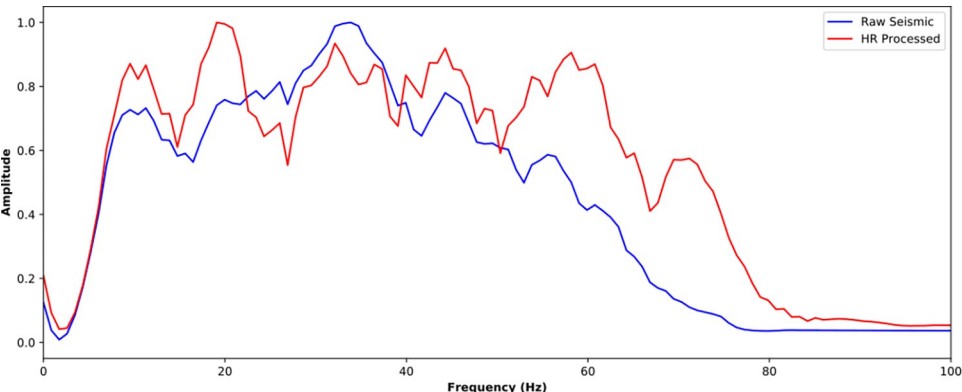

**Fig 21. Spectra comparison of near-well raw trace and its corresponding HR processed trace from the prestack seismic data.**

good validation performance, excavating the mapping relationship between LR and HR seismic. Applications to raw poststack and prestack seismic data demonstrates that the proposed method effectively improves resolution and maintains the continuity of seismic events. Moreover, the HR processing speed is fast thanks to the 1D network architecture. In conclusion, the proposed SCNN model provides a feasible method for HR seismic processing. To make better performance, more works could be done to prepare the dataset, such as incorporating more geological models or logging curves and convolving them with more different LR and HR wavelets to extend the practicality and completeness of the training dataset. Furthermore, adjusting the network's architecture or adopting other optimizers and regularization techniques may improve the accuracy and generalization ability of the proposed deep learning model.

## Supporting information

**S1 File. LR Seismic synthetic by convolution from 14 wells with a 30Hz Richer wavelet.**
(SGY)

**S2 File. HR Seismic synthetic by convolution from 14 wells with a wide-band Ricker wavelet.**
(SGY)

**S3 File. LR Seismic synthetic by convolution from Marmousi2 model with a 30Hz Richer wavelet.**
(SEGY)

**S4 File. HR Seismic synthetic by convolution from Marmousi2 model with a wide-band B-spline wavelet.**
(SEGY)

**S5 File. Training dataset for HR seismic processing SCNN model from well loggings and Marmousi2 model.**
(H5)

**S6 File. HR prediction by SCNN model from LR seismic synthetic of Marmousi2 model.**
(SEGY)

## Acknowledgments

The authors acknowledge the data provided by PST Service Corporation, we also thank J. Li and H. Huang for valuable discussions.

## Author Contributions

**Conceptualization:** Zhenyu Yuan, Yong Wang.

**Data curation:** Zhenyu Yuan, Yong Wang.

**Funding acquisition:** Zheli An.

**Investigation:** Yong Wang.

**Methodology:** Zhenyu Yuan, Zheli An.

**Project administration:** Yuxin Jiang, Weibin Ma.

**Resources:** Yuxin Jiang, Weibin Ma.

**Supervision:** Yuxin Jiang, Weibin Ma.

**Validation:** Zheli An.

**Visualization:** Zhenyu Yuan.

**Writing – original draft:** Zhenyu Yuan, Zheli An, Yong Wang.

**Writing – review & editing:** Zhenyu Yuan.

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
