## [Decision Letter · Decision Letter 0]

15 Aug 2023

PONE-D-23-14881Seismic Resolution Improving by a Sequential Convolutional Neural NetworkPLOS ONE

Dear Dr. YUAN,

Thank you for submitting your manuscript to PLOS ONE. After careful consideration, we feel that it has merit but does not fully meet PLOS ONE’s publication criteria as it currently stands. Therefore, we invite you to submit a revised version of the manuscript that addresses the points raised during the review process.

Even though PLoS One accepts scientifically rigorous research regardless of novelty, the concerns raised by one of the reviewers (similarity to recently published paper) needs to be proper addressed. Besides, some methodological aspects and limitations pointed by the other reviewer need to be analyzed, especially the ones regarding a "inverse crime" situation and the correlation between training and validation/test data. For final acceptance, it should be demonstrated that the results would be interesting to the readers, showing the applicability of the methods to other datasets etc. Please carefully update the paper to account for that.

We look forward to receiving your revised manuscript.

Kind regards,

Luan Carlos de Sena Monteiro Ozelim, D.Sc.

Academic Editor

PLOS ONE

Please share the data either with the manuscript as a supplementary file, or in a repository, and share the link in the Data availability statement.

“This work was supported in part by the National Natural Science Foundation of China High-speed Rail Joint Fund under Grant U1934218.”

Additional Editor Comments:

Dear Authors,

Even though PLoS One accepts scientifically rigorous research regardless of novelty, the concerns raised by one of the reviewers (similarity to recently published paper) needs to be proper addressed. It should be demonstrated that the results would be interesting to the readers, showing the applicability of the methods to other datasets etc.

Please carefully update the paper to account for that (show the differences, if any, or discuss novel and interesting applications of the methods)

Reviewers' comments:

Reviewer's Responses to Questions

**Comments to the Author**

1. Is the manuscript technically sound, and do the data support the conclusions?

Reviewer #1: Yes

Reviewer #2: Yes

2. Has the statistical analysis been performed appropriately and rigorously? 

Reviewer #1: N/A

Reviewer #2: No

3. Have the authors made all data underlying the findings in their manuscript fully available?

Reviewer #1: No

Reviewer #2: No

4. Is the manuscript presented in an intelligible fashion and written in standard English?

Reviewer #1: Yes

Reviewer #2: Yes

5. Review Comments to the Author

Reviewer #1: Well-written paper, although it needs another round of proofreading to fix typos/errors (examples: "closely", not "closed" correlated in 2.1; "Ricky" wavelet in 2.2; reference error after Fig 2).

If re-submitted, I think it would benefit from a few clarifications/additions:

1) You note that CNN's can more effectively use spatial information present in seismic images, but your method is still 1D. Why not use a 2- or 3-D method like some of the examples you cited?

2) In 2.2, explain why using analytical processing techniques to obtain HR training data "cannot have good applicability".

3) The entire discussion on wavelets seems to be of limited relevance to the overall paper, especially since we can't "choose" our wavelet in the field. Consider condensing it.

4) Consider adding a discussion on the limitations of convolutional modeling for applications like this, especially since your Marmousi "test" examples were presumably obtained with the same wavelet/convolutional process as the training data used to obtain the model. This is close to "inverse crime" - your results might say more about the CNN's ability to learn your particular convolution/deconvolution operator than its ability to improve real seismic signals.

5) Would appreciate better quantification of your models' performance, especially for the well-trained examples like Fig 10. For example, show improvement in correlation between HR prediction/HR synthetic vs LR synthetic/HR synthetic.

6) The "perfect" Marmousi results in Fig 11-13 suggest that your training and validation/test data are highly correlated (along with potential issues mentioned in point 4 above). Instead of randomly assigning traces as training/validation, it might be more instructive to partition the image to ensure that the validation traces are very far (>>20 traces) away from training data.

7) Thank you for including the spectral analysis of the post- and pre-stack examples. Since there is a pretty clear "band" of improvement in the HR predictions (~40-60 Hz in both examples), it would be very interesting to bandpass the processed images to that range and show the signals that have been enhanced by your method (and ensure that we are not simply adding noise at those frequencies).

Thanks for the interesting work.

Reviewer #2: The paper presents an approach for detecting thin-layer rocks. However, improving resolution in seismic signal has a long-standing history and a lot of methods has been devised for this purpose.

This approach, which is based on sequential convolution network (autoencoder type of structure) with training on the Marmousi2 Model, is similar to one of recent published paper.

Iqbal, N. (2022). DeepSeg: Deep segmental denoising neural network for seismic data. IEEE Transactions on Neural Networks and Learning Systems.

In this regards, the authors need to provide their contributions and novelty.

Is the training generalized and the network can be tested on nay dataset? If that is the case, more examples may be needed.

6. PLOS authors have the option to publish the peer review history of their article (what does this mean?). If published, this will include your full peer review and any attached files.

Reviewer #1: No

Reviewer #2: No

---

## [Author Response · Author response to Decision Letter 0]

12 Mar 2024

Reviewer #1: Well-written paper, although it needs another round of proofreading to fix typos/errors (examples: "closely", not "closed" correlated in 2.1; "Ricky" wavelet in 2.2; reference error after Fig 2).

A: Thank you for pointing out the spelling error, we have proofread the manuscript roundly and fixed some typos, and adjusted some expressions to make the manuscript concise and smooth.

If re-submitted, I think it would benefit from a few clarifications/additions:

1) You note that CNN's can more effectively use spatial information present in seismic images, but your method is still 1D. Why not use a 2- or 3-D method like some of the examples you cited?

A: The application of 1D CNN architecture has two advantages: (1) the processing speed of 1-D method is faster than 2- or 3-D method with considering the spatial information along depth; (2) not limited by the lateral variations of the poststack or different prestack seismic data, suitable for processing both pre-stack and post-stack data.

2) In 2.2, explain why using analytical processing techniques to obtain HR training data "cannot have good applicability".

A: Sorry for the absolute expression, we have revised the expression to “which may lack wide suitability” since the analytical processing techniques are usually applicable to specific conditions.

3) The entire discussion on wavelets seems to be of limited relevance to the overall paper, especially since we can't "choose" our wavelet in the field. Consider condensing it.

A: We appreciate your comments on our research design. Considering wavelets’ contribution to the training dataset generation, we have adjusted the structure of the manuscript after discussing with the rest of the authors. The former chapter 2 was spilt into two chapters, named “2. Sequential convolutional neural network” and “3. Data preparation”, where the discussion on wavelets was set as a separate section “3.1 Wavelet selection” to discuss their contribution to dataset generation. Limited by current research, only theoretical wavelets are discussed in this section. However, more different wavelets, contributing to the practicality and completeness of the training dataset, need to be included in follow-up studies, which we have mentioned in the chapter “6. Conclusions”.

4) Consider adding a discussion on the limitations of convolutional modeling for applications like this, especially since your Marmousi "test" examples were presumably obtained with the same wavelet/convolutional process as the training data used to obtain the model. This is close to "inverse crime" - your results might say more about the CNN's ability to learn your particular convolution/deconvolution operator than its ability to improve real seismic signals.

A: Thanks for your professional comments. For the “inverse crime” problem, we prepared the training dataset from not only Marmousi2 model but also well loggings, and the HR synthetics are convolved by the wide-band B-spline wavelet and wide-band Ricker wavelet separately. The above operation may mitigate the so-called “inverse crime” problem to some extent. What’s more, we have rewritten the chapter “6. Conclusions” and added some discussions, including “To make better performance, more works could be done to prepare the dataset, such as incorporating more geological models or logging curves and convolving them with more different LR and HR wavelets to extend the practicality and completeness of the training dataset.”

5) Would appreciate better quantification of your models' performance, especially for the well-trained examples like Fig 10. For example, show improvement in correlation between HR prediction/HR synthetic vs LR synthetic/HR synthetic.

A: For Fig 10, the well logging case, Pearson correlation coefficients are calculated for HR prediction/HR synthetic and LR synthetic/HR synthetic as 0.42 and 0.35. However, for Fig 11, the Marmousi2 model case, the HR prediction is almost same with the HR synthetic. In summary, though the proposed CNN model performs well for theoretical model, there is considerable room for improvement to apply to more complex geological conditions.

6) The "perfect" Marmousi results in Fig 11-13 suggest that your training and validation/test data are highly correlated (along with potential issues mentioned in point 4 above). Instead of randomly assigning traces as training/validation, it might be more instructive to partition the image to ensure that the validation traces are very far (>>20 traces) away from training data.

A: Thanks for your constructive suggestion. After carefully considering this comment along with the issues you mentioned in point 4, we do think there is lots of tuning work to do to further improve the network’s performance and we would like to do it in the near future. Hope to get your understanding, thanks.

7) Thank you for including the spectral analysis of the post- and pre-stack examples. Since there is a pretty clear "band" of improvement in the HR predictions (~40-60 Hz in both examples), it would be very interesting to bandpass the processed images to that range and show the signals that have been enhanced by your method (and ensure that we are not simply adding noise at those frequencies).

A: For poststack seismic HR processing, the bandpass filtering (40Hz~70Hz) was performed to the HR processed seismic. The filtered image (showed as follow) indicates that the proposed HR processing method adds some high-frequency components to reveal more reflection details. 

Thanks for the interesting work.

Reviewer #2: The paper presents an approach for detecting thin-layer rocks. However, improving resolution in seismic signal has a long-standing history and a lot of methods has been devised for this purpose.

This approach, which is based on sequential convolution network (autoencoder type of structure) with training on the Marmousi2 Model, is similar to one of recent published paper.

Iqbal, N. (2022). DeepSeg: Deep segmental denoising neural network for seismic data. IEEE Transactions on Neural Networks and Learning Systems.

In this regards, the authors need to provide their contributions and novelty.

A: There are some similarities between the “DeepSeg” paper and this manuscript: (1) Seismic trace is taken as processing object, (2) wavelets are adopted to generate training dataset.

Otherwise, we think there are more differences, including (1) the proposed method is for HR processing, while the “DeepSeg” paper is for SNR enhancing, (2) correspondingly, HR and LR wavelets are used to generate HR and LR seismic trace pairs in this manuscript, while in the “DeepSeg” paper, wavelets are used to generate high SNR data and noise is added to generate the low SNR data, (3) the time-frequency transform performed in the “DeepSeg” paper is good for extracting more features in frequency domain and provides a referential idea for our research, meanwhile the SCNN architecture adopted in our manuscript has advantage of fast processing speed, (4) both well loggings and geological models are utilized to generate training dataset, considering some practical geological conditions.

What’s more, the proposed method is a continuation of the authors’ former conference paper (Yuan et al. 2021. Improving Seismic Resolution by a Sequential Convolutional Neural Network. 82nd EAGE Annual Conference & Exhibition).

In summary, this manuscript contributes to CNN-based seismic HR processing deep learning and shows some advantages in fast and reliable HR processing for both poststack and prestack seismic data.

Is the training generalized and the network can be tested on nay dataset? If that is the case, more examples may be needed.

A: For model training, both well loggings and Marmousi2 model (extracting traces with interval of 20) are utilized to generate training dataset to simulate practical geologic formations. Furthermore, the trained model was applied to the whole Marmousi2 model and practical poststack and prestack seismic data to achieve appreciable HR processing performance. Admittedly, there is still room for the proposed HR processing neural network to improve, we have discussed this issue in chapter “6. Conclusions” (“To make better performance, more works could be done to prepare the dataset, such as incorporating more geological models or logging curves and convolving them with more different LR and HR wavelets to extend the practicality and completeness of the training dataset. Furthermore, adjusting the network’s architecture or adopting other optimizers and regularization techniques may improve the accuracy and generalization ability of the proposed deep learning model.”) and will carry on this improvement in our future work.

---

## [Decision Letter · Decision Letter 1]

5 Apr 2024

PONE-D-23-14881R1Seismic Resolution Improving by a Sequential Convolutional Neural NetworkPLOS ONE

Dear Dr. YUAN,

Thank you for submitting your manuscript to PLOS ONE. After careful consideration, we feel that it has merit but does not fully meet PLOS ONE’s publication criteria as it currently stands. Therefore, we invite you to submit a revised version of the manuscript that addresses the points raised during the review process.

In special, the authors must indicate how their study differs from previously published ones (especially their own research). Not only that, they must indicate why their study is of interest to the readers of the academic community. Addressing these two issues is mandatory to further evaluate the paper.

We look forward to receiving your revised manuscript.

Kind regards,

Luan Carlos de Sena Monteiro Ozelim, D.Sc.

Academic Editor

PLOS ONE

Journal Requirements:

Thank you for your response to our previous query regarding the PLOS ONE requirements for code sharing. Please note that we expect all researchers with submissions to PLOS in which author-generated code underpins the findings in the manuscript to make all author-generated code available without restrictions upon publication of the work. In cases where code is central to the manuscript, we may require the code to be made available as a condition of publication. Authors are responsible for ensuring that the code is reusable and well documented (https://journals.plos.org/plosone/s/materials-software-and-code-sharing#loc-sharing-code).

You have indicated that the code underpinning this work may be made available depending on further negotiation with PST Service Corporation. Please therefore discuss this requirement and ensure that the code is made available with your revised manuscript files, using one of the recommended methods provided on the page linked above. Please note that if this requirement cannot be met, we may reject your submission.

Reviewers' comments:

Reviewer's Responses to Questions

**Comments to the Author**

1. If the authors have adequately addressed your comments raised in a previous round of review and you feel that this manuscript is now acceptable for publication, you may indicate that here to bypass the “Comments to the Author” section, enter your conflict of interest statement in the “Confidential to Editor” section, and submit your "Accept" recommendation.

Reviewer #1: All comments have been addressed

Reviewer #2: (No Response)

2. Is the manuscript technically sound, and do the data support the conclusions?

Reviewer #1: Yes

Reviewer #2: (No Response)

3. Has the statistical analysis been performed appropriately and rigorously? 

Reviewer #1: N/A

Reviewer #2: Yes

4. Have the authors made all data underlying the findings in their manuscript fully available?

Reviewer #1: No

Reviewer #2: No

5. Is the manuscript presented in an intelligible fashion and written in standard English?

Reviewer #1: Yes

Reviewer #2: Yes

6. Review Comments to the Author

Reviewer #1: (No Response)

Reviewer #2: 1) What is the difference or contribution of the authors as compared to the conference paper, Yuan

et al. 2021. Improving Seismic Resolution by a Sequential Convolutional Neural Network. 82nd EAGE

Annual Conference & Exhibition

2)The responses need to be highlighted in the revised manuscript.

7. PLOS authors have the option to publish the peer review history of their article (what does this mean?). If published, this will include your full peer review and any attached files.

Reviewer #1: No

Reviewer #2: No

---

## [Author Response · Author response to Decision Letter 1]

26 Apr 2024

Comment 1: What is the difference or contribution of the authors as compared to the conference paper, Yuan et al. 2021. Improving Seismic Resolution by a Sequential Convolutional Neural Network. 82nd EAGE Annual Conference & Exhibition

Response 1: The conference paper is an effectiveness inquiry of sequential convolutional neural network (SCNN) model applied in seismic resolution processing, this manuscript is a development of the cited conference paper, and there are some big differences. It is known that the quality and size of the training dataset are crucial to the effectiveness of deep learning models. In the cited conference paper, input and output dataset were generated from 14 well loggings by Ricker wavelet and wide-band Ricker wavelet. However, in this manuscript, there different wavelets were analyzed in section “3.1 Wavelet selection”, the wide-band B-spline wavelet exhibited the best ability for object distinction and was chosen to generate high-resolution (HR) seismic traces. In addition to the 14 wells, the Marmousi2 model containing more than 13,000 traces was adopted to generate low-resolution and high-resolution seismic traces, providing a large dataset for HR seismic processing deep learning. Base on the new training dataset, the HR processing SCNN model achieved an improved training and evaluation performance. Furthermore, the trained model was applied to both raw poststack and prestack seismic data from one practical survey, demonstrating good performance in resolution improvement and seismic events continuity maintenance.

Comment 2: The authors must indicate why their study is of interest to the readers of the academic community

Response 2: This manuscript proposed a sequential convolutional neural network (SCNN) model to perform seismic high-resolution (HR) processing. There are some advantages: (1) the network architecture takes advantage of encoder-decoder network and residual learning, is appropriate for learning complex transformation between LR and HR trace pairs. (2) the adoption of 1D CNNs has advantage of fast processing speed than 2- or 3-D networks with considering the spatial information along depth, and is not limited by the lateral variations of the poststack or different prestack seismic data, suitable for processing both pre-stack and post-stack data; (3) the resolution of different wavelets is analyzed, and both well loggings and geological models are utilized to generate training dataset, considering some practical geological conditions. In summary, this manuscript contributes to CNN-based seismic HR processing deep learning and shows some advantages in fast and reliable HR processing for both poststack and prestack seismic data.

---

## [Editor Report · Decision Letter 2]

22 May 2024

Seismic Resolution Improving by a Sequential Convolutional Neural Network

PONE-D-23-14881R2

Dear Dr. YUAN,

We’re pleased to inform you that your manuscript has been judged scientifically suitable for publication and will be formally accepted for publication once it meets all outstanding technical requirements.

Kind regards,

Luan Carlos de Sena Monteiro Ozelim, D.Sc.

Academic Editor

PLOS ONE

Additional Editor Comments (optional):

After addressing the issues raised, the paper can now be accepted.
---

## [Editor Report · Acceptance letter]

31 May 2024

PONE-D-23-14881R2 

PLOS ONE

Dear Dr. Yuan, 

I'm pleased to inform you that your manuscript has been deemed suitable for publication in PLOS ONE. Congratulations! Your manuscript is now being handed over to our production team.

Kind regards, 

on behalf of

Dr. Luan Carlos de Sena Monteiro Ozelim 

Academic Editor

PLOS ONE